atmospheric chemistry/computer modelling and simulation/chemical engineering

select catalyst reduction (SCR), de-NO$_x$ process, power plants, NO$_x$ emission prediction, delay estimation, dynamic inferential model

**Author for correspondence:**
Ze Dong
e-mail: dongze33@126.com

# Dynamic inferential NO$_X$ emission prediction model with delay estimation for SCR de-NO$_X$ process in coal-fired power plants

Laiqing Yan[1], Ze Dong[2], Hao Jia[1], Jianan Huang[1] and Lei Meng[3]

[1]School of Control and Computer Engineering, North China Electric Power University, Beijing 102206, People's Republic of China
[2]Hebei Technology Innovation Center of Simulation and Optimized Control for Power Generation, North China Electric Power University, Baoding 071003, People's Republic of China
[3]Datang Environment Industry Group Co., Ltd, Beijing 100192, People's Republic of China

LY, 0000-0003-2187-5933; ZD, 0000-0003-3269-2913

The selective catalytic reduction (SCR) decomposition of nitrogen oxide (de-NO$_x$) process in coal-fired power plants not only displays nonlinearity, large inertia and time variation but also a lag in NO$_x$ analysis; hence, it is difficult to obtain an accurate model that can be used to control NH$_3$ injection during changes in the operating state. In this work, a novel dynamic inferential model with delay estimation was proposed for NO$_x$ emission prediction. First, $k$-nearest neighbour mutual information was used to estimate the time delay of the descriptor variables, followed by reconstruction of the phase space of the model data. Second, multi-scale wavelet kernel partial least square was used to improve the prediction ability, and this was followed by verification using benchmark dataset experiments. Finally, the delay time difference method and feedback correction strategy were proposed to deal with the time variation of the SCR de-NO$_x$ process. Through the analysis of the experimental field data in the steady state, the variable state and the NO$_x$ analyser blowback process, the results proved that this dynamic model has high prediction accuracy during state changes and can realize advance prediction of the NO$_x$ emission.

## 1. Introduction

During the operation of coal-fired power plants, NO$_x$ emissions discharged into the atmosphere via the exhaust gas are very

harmful to human health and the environment. Meeting pollutant discharge regulations using traditional combustion control is difficult, so selective catalytic reduction (SCR) systems have been widely installed in the flue for the decomposition of nitrogen oxide (de-$NO_x$) [1]. The efficiency of the SCR de-$NO_x$ process can be easily affected by factors such as $NH_3$ injection, dilution air, reaction temperature and the catalyst activity. It is difficult to ensure the optimal ratio of $NH_3$ to $NO_x$ when the coal feed rate changes and the command of the automatic generation control fluctuates rapidly. The reasons for this are as follows: firstly, the SCR de-$NO_x$ process is nonlinear, has a large inertia and varies with time; secondly, the response of the $NO_x$ analyser has a large time delay of approximately 1 min; thirdly, every 50 min, the $NO_x$ analyser performs a blowback process lasting approximately 5 min. When the measured $NO_x$ emission value, which is maintained by the control processor during blowback, is too high or too low, the action of the proportional–integral–derivative (PID) control generally leads to an imbalance between the $NH_3$ injection and the required $NO_x$ reduction. This results in the $NO_x$ emission suddenly increasing or decreasing after blowback. This work aims to provide a method to predict the $NO_x$ emission in a timely manner through the operating variables in coal-fired power plants.

Many data-driven modelling techniques have recently emerged that established black-box models based on measured data from the SCADA system. Zambrano *et al.* [2] adopted the Hammerstein–Wiener model to optimize $NH_3$ injection. Krijnsen *et al.* [3] used neural networks (NN), nonlinear autoregressive exogenous (NARX) models and polynomial fitting to predict the $NO_x$ emission of a diesel engine. For coal-fired boilers, Peng *et al.* [4] established a hybrid ARX model with Gaussian radial basis function network-style coefficients under the steady state. Safdarnejad *et al.* [5] developed a data-driven model based on the recurrent NN model and the dynamic particle swarm optimizer to simultaneously estimate $NO_x$ and CO emissions. Tuttle *et al.* [6] presented a unique NN model using swappable synapse weights and the hybrid optimization approach in a combustion optimization system. For the SCR de-$NO_x$ process, Si *et al.* [7] used an improved online support vector regression (SVR) technique for modelling. Wu *et al.* [8] used an $NO_x$ emission prediction model that was only related to $NH_3$ injection. However, it would not be able to correctly reflect the other factors that affect the $NO_x$ emissions.

For complex chemical process, the high dimensionality and collinearity of the measured data make modelling difficult. The radial basis function kernel partial least square (RBF-KPLS) model can deal with the high dimensionality and collinearity of data [9]. If the sample features contain heterogeneous information, the use of a single kernel for mapping all the samples is not reasonable. Bao *et al.* [10] used a multi-scale kernel to improve the prediction accuracy of the support vector machine (SVM) model. For industrial process modelling, it is difficult to realize accurate results using the RBF kernel model. Zhang *et al.* [11] proposed the Morlet wavelet kernel SVR, and they verified that it has a smaller prediction error than the RBF kernel SVR via the mathematical function.

Because of the lag associated with $NO_x$ analysers, the determined $NO_x$ emission does not reflect the $NH_3$ flow in real time. The phase space of the model sample can be reconstructed by estimating the descriptor variable's delay time to improve the prediction accuracy [12]. In general, the delay time is estimated by field experiments, so its accuracy is usually low. The mutual information (MI) parameter can be used to analyse linear and nonlinear correlations [13]. For the SCR de-$NO_x$ process, the coal feed rate, inlet flue gas flow and inlet flue gas temperature affect the $NO_x$ emission, and there are interactions between the inlet flue gas flow and the inlet flue gas temperature.

To improve the accuracy of the $NO_x$ emission prediction model and solve the time-varying problem for the SCR de-$NO_x$ process, a novel dynamic inferential model is proposed in this paper. First, the $k$-nearest neighbour MI (knnMI) is used to estimate the time delay and reconstruct the model sample. Then, the model brings the Morlet wavelet kernel, which is able to effectively characterize data variation into a multi-scale KPLS. Finally, the delay time difference (DTD) method is used to update the model and the feedback correction strategy to correct the model.

This paper is organized as follows: the theory of the knnMI estimator and the KPLS model are introduced in §2; §3 describes data preprocessing, delay estimation and model reconstruction, model update and correction approach and the framework of the dynamic inferential model; in §4, to evaluate the accuracy of the multi-scale wavelet kernel partial least square (mwKPLS) predictions, it is compared with the RBF-KPLS, multi-scale RBF-KPLS (mRBF-KPLS), wavelet KPLS (wKPLS), particle swarm optimization back propagation (PSO-BP) and SVR based on cross-validation optimization (CV-SVR) models using benchmark datasets; §5 details the experimental results of the dynamic inferential model for the SCR de-$NO_x$ process; finally, concluding remarks are provided in §6.

# 2. Theory

## 2.1. k-nearest neighbour mutual information estimator

Estimation of MI derives from the concept of entropy in information theory. As a measure of information, it reflects the measure of the statistical dependence between two variables. The basic histogram and kernel estimator that belong to the MI estimator are based on probability density estimation. However, they have weaknesses, such as computational complexity, low precision and large amounts of calculation in higher dimensions. The knnMI estimator avoids the shortcoming of exact probability density estimation, and it is simple and only requires a small amount of calculation, which can be summarized as follows [14].

Suppose a space $Z = (x,y)$; here, the vectors $x$ and $y$ are each formed by 1 column and $n$ samples. The estimate for the MI of vectors $x$ and $y$ is then

$$\mathrm{MI}(x,y) = \psi(k) - \langle \psi(n_x + 1) + \psi(n_y + 1) \rangle + \psi(n), \tag{2.1}$$

where $n_x(i)$ is the number of sample points $x_j$, whose distance from $x_i$ is strictly less than $\varepsilon_i/2$, $\varepsilon_i/2$ is denoted as the distance from $x_i$ to its $k$th neighbour; similarly, $n_y(i)$ is obtained instead of $y$, $i \in [1, \ldots, n]$. $\Psi(x)$ is the digamma function, $\Psi(x) = \Gamma(x)^{-1} \mathrm{d}\Gamma(x)/\mathrm{d}x$. It satisfies the recursion $\Psi(x + 1) = \Psi(x) + 1/x$ and $\Psi(1) \approx -0.5772156$. The symbol $\langle \cdots \rangle$ indicates the mean of the variables in it.

## 2.2. Kernel partial least square model

Assuming the descriptor variable matrix $X \in R^{n \times m}$, response variable vector $Y \in R^{n \times 1}$, $i = 1, 2, \ldots,$ $n$. For the kernel matrix $K_0$, its centralized form is $K_1$. $X$ and $Y$ are z-score normalized as $X^1$ and $Y^1$.

The estimation of the KPLS model from the training set is described as follows [9]:

1. Normalizing the training set $X_{\mathrm{tr}}^0$ and $Y_{\mathrm{tr}}^0$, to get $X_{\mathrm{tr}}^1$ and $Y_{\mathrm{tr}}^1$.
2. Calculating the training kernel matrix

$$K_{\mathrm{tr}}^0 = k(x_{\mathrm{tr}}, x_{\mathrm{tr}}). \tag{2.2}$$

3. Centring the training kernel matrix

$$K_{\mathrm{tr}}^1 = \left(I - \frac{1}{n}\mathbf{1}_n\mathbf{1}_n^{\mathrm{T}}\right)K_{\mathrm{tr}}^0\left(I - \frac{1}{n}\mathbf{1}_n\mathbf{1}_n^{\mathrm{T}}\right), \tag{2.3}$$

where $I$ is a unit matrix; $\mathbf{1}_n$ is a matrix where all the elements are 1 with dimensions of $n$.
4. Let $L$ be the number of principal components, and $i$ iterates from 1 to $L$ and randomly initializes the score vector $u^i$ of $X_{\mathrm{tr}}^1$.
5. Calculate the score vector $t^i$

$$t^i = \frac{K_{\mathrm{tr}}^1 u^i}{\|K_{\mathrm{tr}}^1 u^i\|}. \tag{2.4}$$

6. Calculate the weight vector $c^i$

$$c^i = (Y_{\mathrm{tr}}^1)^{\mathrm{T}} t^i. \tag{2.5}$$

7. Calculate the score vector $u^i$

$$u^i = \frac{Y_{\mathrm{tr}}^i c^i}{\|Y_{\mathrm{tr}}^i c^i\|}. \tag{2.6}$$

8. Then steps (4)–(7) are repeated until $t^i$ converges.
9. The matrices $K_{\mathrm{tr}}^1$ and $Y_{\mathrm{tr}}^1$ are reduced until $t$ and $u$ are extracted.

$$K_{\mathrm{tr}}^{i+1} = [I - t^i(t^i)^{\mathrm{T}}]K_{\mathrm{tr}}^i[I - t^i(t^i)^{\mathrm{T}}] \tag{2.7}$$

and

$$Y_{\mathrm{tr}}^{i+1} = Y_{\mathrm{tr}}^i - t^i(t^i)^{\mathrm{T}} Y_{\mathrm{tr}}^i. \tag{2.8}$$

10. The regression coefficient $\boldsymbol{B}$ is calculated and the regression equation of the training set is obtained.

$$\hat{\boldsymbol{Y}}_{\mathrm{tr}} = \boldsymbol{K}_{\mathrm{tr}}\boldsymbol{B} = \boldsymbol{K}_{\mathrm{tr}}\boldsymbol{U}(\boldsymbol{T}^{\mathrm{T}}\boldsymbol{K}_{\mathrm{tr}}\boldsymbol{U})^{-1}\boldsymbol{T}^{\mathrm{T}}\boldsymbol{Y}_{\mathrm{tr}}, \tag{2.9}$$

where $\boldsymbol{T}$ and $\boldsymbol{U}$ are matrices that are composed of score vectors $\boldsymbol{t}$ and $\boldsymbol{u}$.

The prediction for the test set by the KPLS model is similar to the training set, except for computation of the test kernel matrix $\boldsymbol{K}_{\mathrm{te}}^{0}$ and the centralization of $\boldsymbol{K}_{\mathrm{te}}^{0}$

$$\boldsymbol{K}_{\mathrm{te}}^{0} = k(x_{\mathrm{te}}, x_{\mathrm{tr}}) \tag{2.10}$$

and

$$\boldsymbol{K}_{\mathrm{te}}^{1} = \left(\boldsymbol{K}_{\mathrm{tr}}^{0} - \frac{1}{n}\mathbf{1}_{\mathrm{nt}}\mathbf{1}_{\mathrm{n}}^{\mathrm{T}}\boldsymbol{K}_{\mathrm{te}}^{0}\right)\left(\boldsymbol{I} - \frac{1}{n}\mathbf{1}_{n}\mathbf{1}_{n}^{\mathrm{T}}\right), \tag{2.11}$$

where nt is the number of the test set.

# 3. Dynamic inferential model with delay estimation

## 3.1. Data preprocessing

Data preprocessing includes outlier eliminating and data filtering, which are useful for building a stable model structure.

In this paper, the Pauta criterion was used to eliminate outliers. The formula for this is

$$|x_t - \bar{x}_t| \geq 3\sigma_t, \tag{3.1}$$

where $x_t$ is the suspected outlier at time $t$, $\bar{x}_t$ is the sample mean at time $t$ and $\sigma_t$ is the standard deviation of the sample at time $t$. If the above equation is satisfied, the outlier can be eliminated and replaced with the value of the linear interpolation.

To realize dynamic elimination of outliers, $\bar{x}_t$ and $\sigma_t$ in equation (3.1) used the following equations [15]:

$$\bar{x}_{t+1} = \frac{n-1}{n}\bar{x}_t + \frac{1}{n}x_{t+1} \tag{3.2}$$

and

$$\sigma_{t+1} = \sqrt{\frac{n-2}{n-1}\sigma_t^2 + \frac{1}{n-1}(x_{t+1} - \bar{x}_{t+1})^2}, \tag{3.3}$$

where $n$ is the sample size.

In addition, the Butterworth filter was used to filter data.

## 3.2. Delay estimation and model samples reconstruction

Because the time delay between each set variable vector $x_{\cdot i}$ and the response variable vector $y$ is different, the phase space for each $x_{\cdot i}$ is reconstructed by inserting a different time delay $\tau_i \in [\min(\tau_i), \max(\tau_i)]$ ($\min(\tau_i)$ and $\max(\tau_i)$ are determined by field measurements).

The MI value is related to the dimension $w$ of $x'_{\cdot i}$. A suitable $w$ should cover the most relevant data of $x'_{\cdot i}$ with $y$. Hence, the delay time $\tau_i$ and dimension $w$ at time $t$ are calculated as

$$\max_{\tau_i = \tau'_i, w_i = w'_i} \mathrm{MI}([x_{\cdot i}(t - \tau_i - w_i + 1), \ldots, x_{\cdot i}(t - \tau_i - 1), x_{\cdot i}(t - \tau_i)]^{\mathrm{T}}, [y(t - w_i + 1), \ldots, y(t - 1), y(t)]^{\mathrm{T}})$$
$$\mathrm{s.t.}\min(\tau_i) \leq \tau_i \leq \max(\tau_i), \tau_i + w_i < T_{\max}; i \in [1, m], \tag{3.4}$$

where $T_{\max}$ is the maximum reaction time of the SCR process. The above equation is a constrained multi-variable nonlinear optimization problem. For $m$ set variables, there are $2m$ variables that need to be optimized. Thus, within the scope of the above constraints, a global searcher based on a PSO algorithm maximizes the objective function, thereby obtaining an optimal $\tau' = [\tau'_1, \tau'_2, \ldots, \tau'_m]$ and $w' = [w'_1, w'_2, \ldots, w'_m]$.

By estimating the time delay $\tau'_i$ of each set variable vector $x_{\cdot i}$, the reconstructed descriptor variable matrix $X_{rc}$ is assumed as follows:

$$X_{rc} = \begin{bmatrix} x_{\cdot 1}(t - \tau'_1 - n + 1) & \cdots & x_{\cdot i}(t - \tau'_i - n + 1) \\ \vdots & \ddots & \vdots \\ x_{\cdot 1}(t - \tau'_1 - 1) & \cdots & x_{\cdot i}(t - \tau'_i - 1) \\ x_{\cdot 1}(t - \tau'_1) & \cdots & x_{\cdot i}(t - \tau'_i) \end{bmatrix}. \tag{3.5}$$

## 3.3. Multi-scale wavelet kernel partial least square

The Morlet wavelet kernel adopted in this paper has a strong capability for characterizing data variation that can be used to construct the allowable multi-dimensional tensor product wavelet kernel. The mother function is

$$\psi(x) = \cos(1.75x)\exp\left(-\frac{x^2}{2}\right). \tag{3.6}$$

To prove that the Morlet mother wavelet kernel is an admissible support vector kernel, the following definitions are first introduced.

**Definition (3.1).** (Mercer's condition [16]) *In a double infinite dimensional square integrable space $L_2(\Omega)$, the necessary condition for the kernel $k(x, z)$ that can realize the dot product in a feature space for: $\forall \varphi(x) \neq 0$, $\int \varphi(x)\,dx < \infty$ and $\int\int k(x,z)\varphi(x)\varphi(z)\,dxdz > 0$.*

**Definition (3.2).** (Fourier condition [16]) *If the Fourier transform $F[k](\omega) = (2\pi)^{-N/2}\int_x e^{-i\langle\omega,x\rangle}k(x)\,dx \geq 0$, a translation invariant kernel $k(x, z) = k(x - z)$ is a positive definite kernel, $x, z \in R^N$.*

**Definition (3.3).** (Wavelet kernel satisfying translation invariance [17]) *If $\psi(x)$ is a mother wavelet function, $a$ is a scale parameter, $a > 0$, $b_i$ and $b'_i$ are translation parameters, $b_i, b'_i, x_i \in R, i = 1, 2, \ldots, N, x, z \in R^N$. The wavelet kernel is represented by the dot product as*

$$k(x,z) = \prod_{i=1}^{N}\left[\psi\left(\frac{x_i - b_i}{a}\right)\psi\left(\frac{z_i - b'_i}{a}\right)\right]. \tag{3.7}$$

The tensor product wavelet kernel that satisfies the translation invariance theorem according to definition (3.2) is expressed as

$$k(x,z) = \prod_{i=1}^{N}\left[\psi\left(\frac{x_i - z_i}{a}\right)\right]. \tag{3.8}$$

**Theorem (3.1).** The *Morlet wavelet kernel function satisfies the positive definite condition of the Mercer kernel.*

*Proof.* According to definition (3.1) and equation (3.7), let $\varphi(x) \in R$ and $\varphi(x) \neq 0$, hence

$$F = \iint_{R^N \times R^N} k(x,z)\varphi(x)\varphi(z)\,dxdz$$

$$= \iint_{R^N \times R^N}\prod_{i=1}^{N}\left[\psi\left(\frac{x_i - b_i}{a}\right)\psi\left(\frac{z'_i - b'_i}{a}\right)\right] \times \varphi(x)\varphi(z)\,dxdz$$

$$= \iint_{R^N \times R^N}\prod_{i=1}^{N}\cos\left(1.75\frac{x_i - b_i}{a}\right)\exp\left[-\frac{(x_i - b_i)^2}{2a^2}\right] \times \cos\left(1.75\frac{z'_i - b'_i}{a}\right)\exp\left[-\frac{(z'_i - b'_i)^2}{2a^2}\right]\varphi(x)\varphi(z)\,dxdz$$

$$= \left\{\int_{R^N}\prod_{i=1}^{N}\cos\left(1.75\frac{x_i - b_i}{a}\right)\exp\left[-\frac{(x_i - b_i)^2}{2a^2}\right]\varphi(x)\,dx\right\}^2.$$

Because $\varphi(x) \neq 0$, $F > 0$ can be obtained, therefore, the Mercer's condition is satisfied. ∎

**Theorem (3.2).** *On the basis of the Morlet wavelet kernel, the translation invariant wavelet kernel $k(x) = \prod_{i=1}^{N}\cos\left(1.75\frac{x_i}{a}\right)\exp\left(-\frac{x_i^2}{2a^2}\right)$ is a permissible support vector kernel that is represented by equation (3.7) in definition (3.3).*

*Proof.* Fourier transform for $k(x)$

$$F[k](\omega) = (2\pi)^{-N/2} \int_{R^N} e^{-j\langle\omega,x\rangle} k(x)\, dx$$

$$= (2\pi)^{-N.2} \int_{R^N} e^{-j\langle\omega,x\rangle} \prod_{i=1}^{N} \cos\left(1.75\frac{x_i}{a}\right) \exp\left(-\frac{x_i^2}{2a^2}\right) dx$$

$$= \prod_{i=1}^{N} \frac{1}{\sqrt{2\pi}} \int_{-\infty}^{+\infty} e^{-j\omega_i x_i} \cos\left(\frac{1.75x_i}{a}\right) \exp\left(-\frac{x_i^2}{2a^2}\right) dx$$

$$= \prod_{i=1}^{N} \frac{1}{2\sqrt{2\pi}} \int_{-\infty}^{+\infty} e^{-j\omega_i x_i} \left[\exp\left(j\frac{1.75x_i}{a}\right) + \exp\left(-j\frac{1.75x_i}{a}\right)\right] \exp\left(-\frac{x_i^2}{2a^2}\right) dx$$

$$= \prod_{i=1}^{N} a \int_{-\infty}^{+\infty} \left(e^{j\frac{1.75x_i}{a} - j\omega_i x_i} + e^{-j\frac{1.75x_i}{a} + j\omega_i x_i}\right) dx$$

$$= \prod_{i=1}^{N} \frac{a}{2} \left\{\exp\left[\frac{(1.75 - \omega_i a)^2}{2}\right] + \exp\left[-\frac{(1.75 + \omega_i a)^2}{2}\right]\right\}.$$

Because $a > 0$ and $N > 1$, then, $F > 0$. According to definition (3.3), the Morlet wavelet kernel is a permissible support vector kernel.

The multi-scale kernel takes into account the distribution characteristics of the samples in the original input space. Therefore, it improves the sparsity of the solution in the high-dimensional feature space. Based on the Morlet wavelet kernel, the multi-scale wavelet kernel is represented by

$$k(x,z) = k_1(x,z) + k_2(x,z) + \cdots + k_c(x,z)$$
$$k_c(x,z) = \prod_{i=1}^{N} \left\{\cos\left(1.75\frac{x_i - z_i}{a_c}\right) \exp\left[-\frac{(x_i - z_i)^2}{2a_c^2}\right]\right\} \tag{3.9}$$

where $c$ is the scale parameter, $a_i$ is the wavelet kernel width and $i = 1, \ldots, c$.

To prove that the multi-scale wavelet kernel preserves the finitely positive semi-definite 'kernel' property, the following theorems are introduced. ∎

**Theorem (3.3).** *Kernel matrix is a positive semi-definite matrix.*

*Proof.* Let kernel matrix $\boldsymbol{K} = k(x_i, x_j) = \langle\psi(x_i), \psi(x_j)\rangle$ and $i, j = 1, \ldots, n$. Thus, any vector $\alpha \in R^n$ satisfies:

$$\alpha^{\mathrm{T}} \boldsymbol{K} \alpha = \sum_{i,j=1}^{n} \alpha(i)\alpha(j)\boldsymbol{K} = \sum_{i,j=1}^{n} \alpha(i)\alpha(j)\langle\psi(x_i),\psi(x_j)\rangle = \left\langle \sum_{i=1}^{n} \alpha(i)\psi(x_i), \sum_{i=1}^{n} \alpha(j)\psi(x_j) \right\rangle$$

$$= \left\| \sum_{i=1}^{n} \alpha(i)\psi(x_i) \right\|^2 \geq 0. \tag{3.10}$$

∎

**Theorem (3.4).** *Multi-scale kernel matrix is a positive semi-definite matrix.*

*Proof.* Let the multi-scale kernel matrix $\boldsymbol{K} = \boldsymbol{K}_1 + \boldsymbol{K}_2 + \cdots + \boldsymbol{K}_c = k_1(x_i, x_j) + k_2(x_i, x_j) + \cdots + k_c(x_i, x_j)$, $i, j = 1, \cdots, n$. According to theorem (3.3), any vector $\alpha \in R^n$ satisfies

$$\alpha^{\mathrm{T}} \boldsymbol{K} \alpha = \alpha^{\mathrm{T}} \boldsymbol{K}_1 \alpha + \alpha^{\mathrm{T}} \boldsymbol{K}_2 \alpha + \cdots + \alpha^{\mathrm{T}} \boldsymbol{K}_c \alpha \geq 0. \tag{3.11}$$

∎

Hence, the multi-scale kernel matrix $K$ is positive semi-definite.

A kernel function with a certain kernel width is suitable for mapping a learning sample with a certain feature into a high-dimensional feature space; hence, the feature distribution number can be used as the optimal scale parameter. In this paper, fuzzy $c$-means (FCM) clustering was used to partition the sample feature distribution, so that the optimal classification is selected as the scaling parameter.

If the descriptor variable matrix $X \in R^{n \times m}$ has $c$ cluster centres, the fuzzy classification matrix $\boldsymbol{U}_{c \times n}$ denotes that $n$ samples are partitioned into $c$ classifications. Therefore, in the corresponding cluster centre matrix $\boldsymbol{Z}_{c \times s}$, the $s$th index value is the average of the index value in accordance with the $c$th

classification sample

$$Z_{ij} = \frac{\sum_{l=1}^{n} (U_{il})^2 X_{lj}}{\sum_{l=1}^{n} (U_{il})^2}. \tag{3.12}$$

Then the objective function is constructed

$$J = \sum_{i=1}^{c} \sum_{j=1}^{n} (U_{ij})^2 \|X_j - Z_i\|^2. \tag{3.13}$$

The optimal fuzzy classification matrix $\mathbf{U}$ and the corresponding cluster centre matrix $\mathbf{Z}$ are solved, so that the objective function $J$ reaches a minimum. Here, $\|X_j - Z_i\|$ represents the Euclidean distance between the $j$th sample and the $i$th cluster centre.

The fuzzy classification uncertainty is

$$W_c(U) = \frac{1}{n} \sum_{i=1}^{n} \sum_{j=1}^{c} (U_{ij})^2. \tag{3.14}$$

If equation (3.14) is close to 1, the classification ambiguity is low and the FCM clustering effect is better. When equation (3.14) is at its maximum, that is

$$W_{c^*}(U) = \max_{2 \le c \le n} [\max(W_c(U))]. \tag{3.15}$$

When the above equation is satisfied, the optimal classification is realized for $c^*$.

## 3.4. Dynamic model update method

In this paper, dynamic modelling was realized by increasing the inputs to the model; for this, the historical input $x(t-1), \ldots, x(t-w+1)$ and output $y(t-1), \ldots, y(t-w+1)$ were added as the new input. Furthermore, the time difference (TD) method can solve the variable drift problem and realize improved prediction accuracy compared with other update methods; furthermore, the data-driven model based on the TD method does not require frequent reconstruction and parameter updates [18,19]. The TD method first calculated the first-order TD between adjacent sampling data for the input and output. Here, $\Delta x(t)$ and $\Delta y(t)$ can be calculated as

$$\Delta x(t) = x(t) - x(t-1) \tag{3.16}$$

and

$$\Delta y(t) = y(t) - y(t-1). \tag{3.17}$$

Then, the regression model was expressed as $\Delta y(t) = f[\Delta x(t)]$. After the above regression model was trained, when a new sample $x(t+1)$ is taken, the TD of the input can be calculated as

$$\Delta x(t+1) = x(t+1) - x(t). \tag{3.18}$$

Hence, the TD of the output can be predicted by the training regression model as

$$\Delta y(t+1) = f(\Delta x(t+1)). \tag{3.19}$$

Finally, the actual predicted output was

$$\hat{y}(t+1) = \Delta y(t+1) + y(t). \tag{3.20}$$

In this paper, because of the time-varying SCR de-$NO_x$ process and large time delay for $NO_x$ analysis, the DTD update method and feedback correction strategy are proposed. In §3.2, the model matrix is reconstructed by delay estimation. Therefore, the training regression model becomes

$$\Delta y(t) = f[\Delta x_{.i}(t - \tau'_i)] = f[x_{.i}(t - \tau'_i) - x_{.i}(t - \tau'_i - 1)]. \tag{3.21}$$

Similarly, the DTD of the output can be predicted by:

$$\Delta y(t+1) = f(\Delta x_{.i}(t+1 - \tau'_i)). \tag{3.22}$$

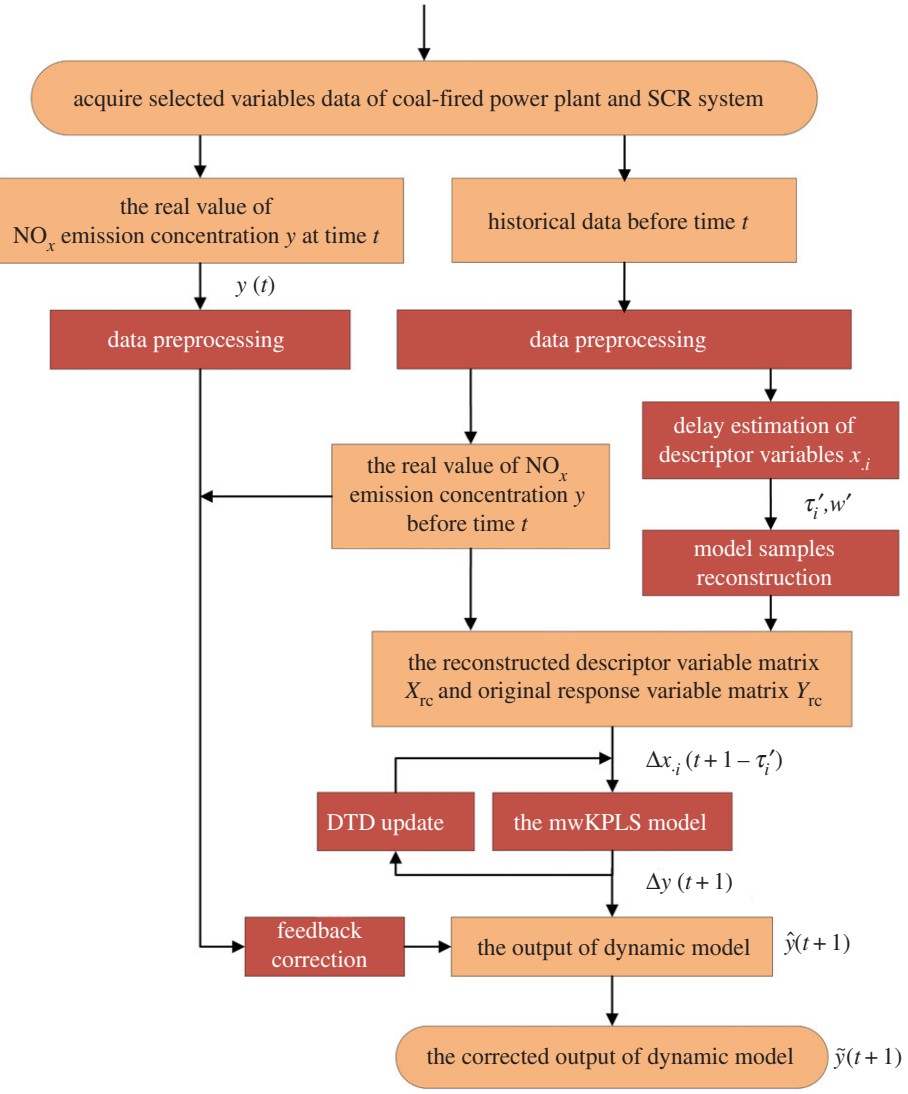

**Figure 1.** Framework for the dynamic inferential model.

Furthermore, the model correction formula to compensate for the prediction error caused by changes of the operating state is as follows:

$$\tilde{y}(t+1) = \hat{y}(t+1) + \Delta y(t+1), \tag{3.23}$$

$$\Delta y(t+1) = \rho \times \Delta y_0(t+1) + (1-\rho) \times \Delta y(t) \tag{3.24}$$

and

$$\Delta y_0(t+1) = y(t) - \hat{y}(t), \tag{3.25}$$

where $\rho$ is 0.3, $\tilde{y}(t)$ is the corrected dynamic model output, $\hat{y}(t)$ is the dynamic model output and $y(t)$ is the real value.

## 3.5. Framework for the dynamic inferential model

The framework for the dynamic inferential model mainly includes data preprocessing, delay estimation, model sample reconstruction, the multi-scale wavelet kernel partial least square (mwKPLS) model, DTD update and feedback correction (figure 1). The steps of the algorithm are as follows:

1. Acquire the measured data for the selected variables before time $t$ and confirm the raw samples.
2. Preprocess the raw samples including eliminating outliers and filtering.
3. Estimate the descriptor variable's delay time $\tau' = [\tau'_1, \tau'_2, \ldots, \tau'_m]$ and reconstruct the samples.
4. Construct the first-order DTD based on the reconstructed descriptor variable matrix $X_{rc}$ and the original response variable matrix $Y$.

**Table 1.** Parameters of the datasets.

| dataset | size | descriptor variable | response variable | training sample | test sample |
|---|---|---|---|---|---|
| concrete slump | $103 \times 8$ | 7 | 1 | 78 | 25 |
| polymer | $61 \times 11$ | 10 | 1 | 41 | 20 |

**Table 2.** Comparison between the dataset experiment results.

| dataset | algorithm | scale parameter | fuzzy classification uncertainty | RMSE value | |
|---|---|---|---|---|---|
| | | | | training | test |
| concrete slump | PSO-BP | — | — | 2.0927 mg m$^{-3}$ | 4.7421 mg m$^{-3}$ |
| | CV-SVR | — | — | 0.1389 mg m$^{-3}$ | 5.4851 mg m$^{-3}$ |
| | RBF-KPLS | $c = 1$ | — | 1.4020 mg m$^{-3}$ | 4.5867 mg m$^{-3}$ |
| | mRBF-KPLS | $c = 2$ | $W_c(U) = 0.7334$ | 1.1793 mg m$^{-3}$ | 4.0792 mg m$^{-3}$ |
| | mRBF-KPLS | $c = 3$ | $W_c(U) = 0.7039$ | 1.2022 mg m$^{-3}$ | 4.1851 mg m$^{-3}$ |
| | wKPLS | $c = 1$ | — | 1.2869 mg m$^{-3}$ | 4.4404 mg m$^{-3}$ |
| | mwKPLS | $c = 2$ | $W_c(U) = 0.7334$ | 1.0306 mg m$^{-3}$ | 3.8684 mg m$^{-3}$ |
| | mwKPLS | $c = 3$ | $W_c(U) = 0.7039$ | 0.6500 mg m$^{-3}$ | 4.1186 mg m$^{-3}$ |
| polymer | PSO-BP | — | — | 0.0486 | 0.3462 |
| | CV-SVR | — | — | 0.0055 | 0.0679 |
| | RBF-KPLS | $c = 1$ | — | 0.0530 | 0.0852 |
| | mRBF-KPLS | $c = 2$ | $W_c(U) = 0.8946$ | 0.0290 | 0.0684 |
| | wKPLS | $c = 1$ | — | 0.0312 | 0.0830 |
| | mwKPLS | $c = 2$ | $W_c(U) = 0.8946$ | 0.0242 | 0.0671 |

5. Carry out FCM clustering on the reconstructed descriptor variable matrix $X_{rc}$ to determine the optimal scale parameter $c^*$.
6. Normalize the training set and carry out estimation using the mwKPLS model.
7. Predict the NO$_x$ emission using the mwKPLS model based on $a_1, \cdots, a_{c^*}$.
8. Acquire the measured data for selected variables at time $t + 1$ and correct the predicted NO$_x$ emission value based on the feedback.
9. Repeat steps 2–8.

# 4. Benchmark dataset experiments

In this paper, two benchmark datasets—the concrete slump dataset [20] and the polymer dataset [21]—were used to verify the prediction ability of the mwKPLS model. The parameters of the datasets are shown in table 1.

The following models were used for comparison with the mwKPLS model: RBF-KPLS, multi-scale RBF-KPLS (mRBF-KPLS), wKPLS, back propagation NN (BP-NN) based on PSO optimization (PSO-BP) and CV-SVR. The 10-fold CV method was adopted for parameter optimization in all the models except PSO-BP. To avoid parameters in a local optimum, a grid search was used to optimize the kernel width under the same search range, and the root mean square error (RMSE) was used as the evaluation index for model accuracy. The results of the experiment and the parameters of the algorithm that were optimized are shown in tables 2 and 3 ($b$ indicates the number of hidden layer nodes, $p$ indicates penalty parameter and $\sigma$ indicates RBF kernel width).

(1) The wKPLS algorithm had a smaller RMSE value than the KPLS one for the training and test sets at the same $c$. The Morlet mother wavelet kernel is nearly orthogonal with the RBF kernel; hence, the fitting and generalizability of the wKPLS algorithm were improved.

**Table 3.** Parameters of algorithm.

| algorithm | scale parameter | principal component | parameter range | optimal parameter | |
|---|---|---|---|---|---|
| | | | | concrete slump | polymer |
| PSO-BP | — | — | $b \in [15, 20]$ | $b = 19$ | $b = 19$ |
| CV-SVR | — | — | $\sigma \in [-10, 10],$ $p \in [-10, 10]$ | $\sigma = -5, p = 5$ | $\sigma = -10, p = 9$ |
| RBF-KPLS | $c = 1$ | $L = 4$ | $\sigma \in [1, 20]$ | $\sigma = 3$ | $\sigma = 2$ |
| mRBF-KPLS | $c = 2$ | $L = 4$ | $\sigma_1, \sigma_2 \in [1, 20]$ | $\sigma_1 = 2, \sigma_2 = 5$ | $\sigma_1 = 1, \sigma_2 = 3$ |
| mRBF-KPLS | $c = 3$ | $L = 4$ | $\sigma_1, \sigma_2, \sigma_3 \in [1, 20]$ | $\sigma_1 = 2, \sigma_2 = 3, \sigma_3 = 10$ | — |
| wKPLS | $c = 1$ | $L = 4$ | $a \in [1, 20]$ | $a = 7$ | $a = 4$ |
| mwKPLS | $c = 2$ | $L = 4$ | $a_1, a_2 \in [1, 20]$ | $a_1 = 4, a_2 = 10$ | $a_1 = 3, a_2 = 11$ |
| mwKPLS | $c = 3$ | $L = 4$ | $a_1, a_2, a_3 \in [1, 20]$ | $a_1 = 2, a_2 = 5, a_3 = 10$ | — |

(2) For the concrete slump dataset, $W_c(U)$ reached a maximum of 0.7334 when $c = 2$. However, when $c = 3$, the RMSE value of the training set decreased, and the RMSE value of the test set increased. This indicated that the optimal $c$ was related to the sample features. If $c$ was too large, the training accuracy of the model could be improved, but it may not improve the generalizability of the model. Therefore, FCM clustering was used to determine the optimal $c$ effectively.

For the polymer dataset, the $c$ determined by the FCM clustering was at most 2. The mwKPLS algorithm had a smaller RMSE value than the wKPLS one for the training and test sets. The prediction accuracy for the mwKPLS algorithm was the highest. This indicated that the Morlet wavelet kernel is suitable for samples with multiple feature distribution.

(3) Compared with the PSO-BP and CV-SVR algorithms, mwKPLS had the highest prediction accuracy. This indicated that the CV-SVR brought unnecessary redundancy or noise into the training model, resulting in the low prediction accuracy of the model. Because many parameters (except $b$) need to be optimized, the output of the PSO-BP model was not necessarily optimal.

# 5. Field data experiment and result analysis

## 5.1. SCR de-$NO_x$ process

In coal-fired power plants, the SCR de-$NO_x$ reaction is carried out in a reactor that is vertically installed between the boiler economizer and the air preheater. When $NH_3$ and air are mixed, the mixed air passing through the ammonia injection grille in the upper part of reactor reacts with the flue gas from the outlet of the economizer under the catalyst and then passes into the air preheater. Finally, the de-$NO_x$ exhaust gas is discharged into the atmosphere through the chimney. The flow chart for the SCR de-$NO_x$ process is shown in figure 2.

## 5.2. The selection of model variables and samples

The $NO_x$ emission is related to many factors, such as $NH_3$ injection, the dilution air volume, the reaction temperature and the catalyst activity. In addition, the boiler load change, coal quality and combustion conditions (such as the $O_2$ content) cause large fluctuations in the inlet $NO_x$ concentration. The selection of the descriptor variables is generally based on the mechanism of the process. Therefore, this paper mainly considers the steps for $NO_x$ formation and the mechanism of the SCR de-$NO_x$ reaction. $NO_x$ in the flue gas is mainly in the form of NO, with a smaller portion of $NO_2$. The main reactions in SCR de-$NO_x$ process are as follows:

$$4NH_3 + 4NO + O_2 = 4N_2 + 6H_2O, \tag{5.1}$$
$$4NH_3 + 6NO = 5N_2 + 6H_2O \tag{5.2}$$
and
$$2NH_3 + NO + NO_2 = 2N_2 + 3H_2O. \tag{5.3}$$

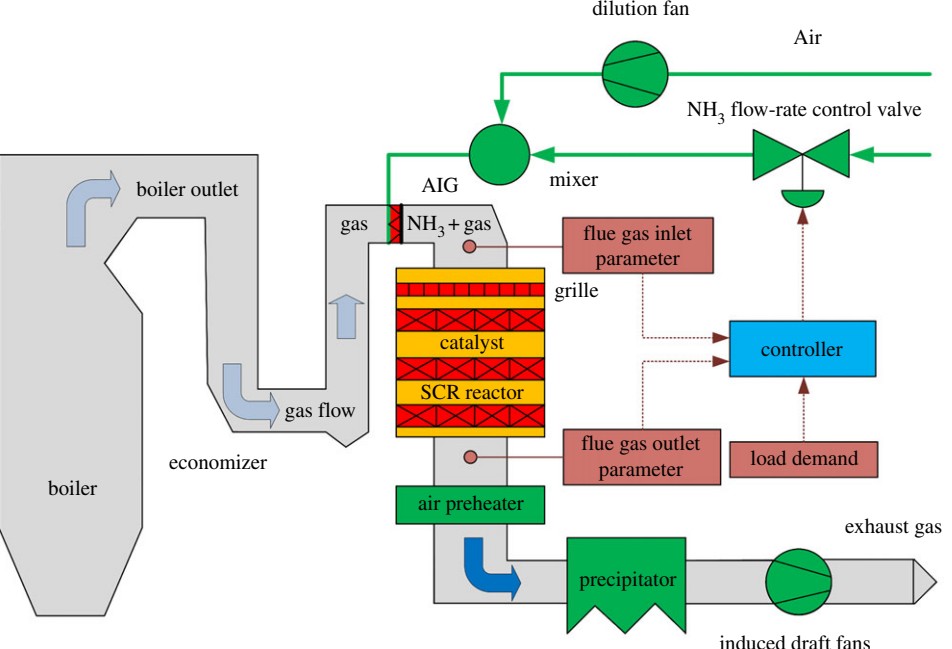

**Figure 2.** The flow chart of the SCR de-NO$_x$ process.

From the above reactions, the inlet NO$_x$ concentration and the NH$_3$ injection flow directly reflect the NH$_3$/NO$_x$ molar ratio that affects the de-NO$_x$ efficiency and the NH$_3$ slip. Furthermore, the SCR reaction is affected by the inlet O$_2$ content. The NH$_3$ injection flow is mainly controlled to adapt for different boiler loads via the NH$_3$ valve. The inlet O$_2$ content directly affects the NO$_x$ emission concentration and de-NO$_x$ efficiency. Further, the boiler load change often affects the inlet flue gas flow, resulting in a change of the flue gas temperature by heat exchange. The change of the inlet flue gas temperature affects the speed of the SCR de-NO$_x$ reaction and the activity of the catalyst.

The experimental field data were continuously recorded in the DCS database of the coal-fired power plant. Assuming that the coal quality was constant, the state of unit covers the steady state and the variable state, in which the load varied between 700 and 900 MW, and the selected data should be continuous. One-dimensional linear interpolation was performed on the measured NO$_x$ emission during the blowback process, and any abnormal operation condition should be avoided. The sampling period was 10 s and a total of 2100 samples were collected. Table 4 shows the range of selected model variables.

## 5.3. Analysis of the data preprocessing results

An assumption of the Pauta criterion is that the data are normally distributed. While the operational data of the practical industrial process rarely conform to a normal distribution, it does not affect the effectiveness of the outlier elimination. The probability that the numerical distribution of industrial process data is within $(\mu - 3\sigma, \mu + 3\sigma)$ is 0.9973. Taking the NO$_x$ emission concentration as an example, outlier elimination was performed using the Pauta criterion. Figure 3$a$ shows that the Pauta criterion was able to detect some obvious outliers, such as data at 100, 250 and 318 min. These outliers were consistently mismatched with the baseline population, which adversely affected the statistical properties of the entire data.

Figure 3$b$ shows that the data before filtration have a large amount of high-frequency noise, which does not help stabilize the model. In this work, the order of the Butterworth filter used was 8 and the cut-off frequency was 0.9. After filtering, the high-frequency noise was eliminated to a large extent, and the filtered data could capture the change of the trend. Therefore, using adaptive filtering to process raw data was beneficial for the predictive model.

## 5.4. Analysis of the delay estimation result

According to the field test results, the maximum delay for the SCR denitration reaction is approximately 120–400 s, and the maximum delay for the boiler load that affects the inlet NO$_x$ concentration is 600 s.

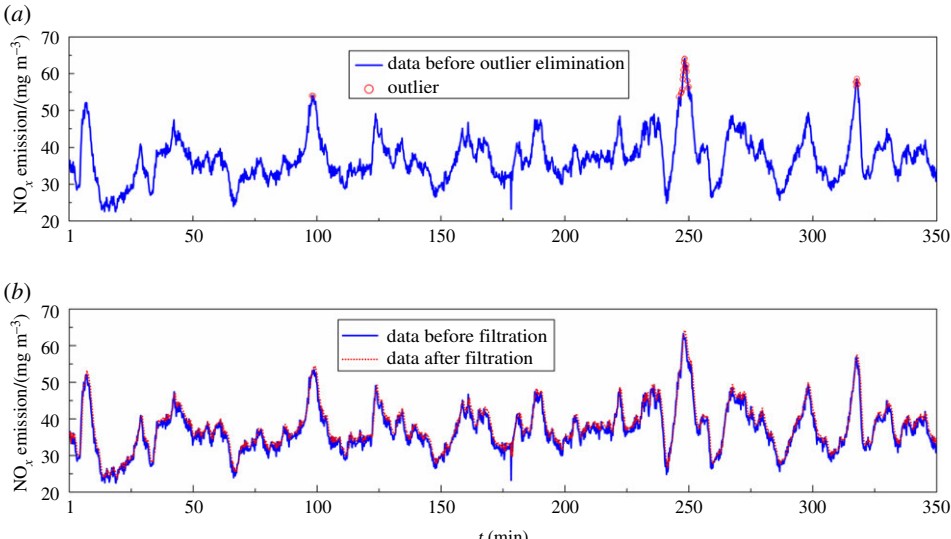

**Figure 3.** The result of outlier elimination and filtration.

**Table 4.** Ranges for selected model variables.

| variable | range |
|---|---|
| boiler load ($N_e$) | 676–898 MW |
| inlet $NO_x$ concentration ($C_{NO_x,in}$) | 127–292 mg m$^{-3}$ |
| inlet $O_2$ content ($C_{O_2}$) | 3.78–5.62% |
| inlet flue gas flow ($F$) | 1220–1630 km$^3$ h$^{-1}$ |
| total coal feed rate ($B$) | 275–382 t h$^{-1}$ |
| $NH_3$ injection flow ($Q$) | 36.76–101.9 kg h$^{-1}$ |
| inlet flue gas temperature ($T$) | 351–370℃ |
| $NO_x$ emission concentration ($C_{NO_x,out}$) | 22–64 mg m$^{-3}$ |

In this work, the sampling period was 10 s and $T_{max}$ in equation (3.4) was 120, and the range for the time delay is shown in table 5. As an example, the delay estimation results for each descriptor variable at time $t = 250$ min are shown in table 5.

## 5.5. Data correlation analysis

To analyse whether the descriptor variable $x_i$ and the response variable $y$ is nonlinear and that there is multi-collinearity between the descriptor variables, correlation analysis was performed on the normalized data. The correlation structures and the Pearson correlation coefficient $|r|$ are shown in figure 4.

It can be seen from figure 4 that $x_i$ and $y$ are nonlinear. Furthermore, $|r| < 0.39$, so there is a low correlation between $x$ and $y$. The unit load, inlet $O_2$ content, inlet flue gas temperature and $NO_x$ emission concentration display very weak correlations, while the inlet $NO_x$ concentration and $NO_x$ emission concentration show a slightly stronger correlation. Therefore, the $NO_x$ emission concentration has a nonlinear relationship with the boiler load and inlet flue gas temperature; hence, the $NO_x$ emission may also increase as the boiler load decreases. In addition, a Pearson correlation coefficient $|r|$ greater than 0.7 was observed between the descriptor variable vectors, including the total coal feed rate, inlet flue gas flow and unit load, inlet $NO_x$ concentration and $NH_3$ injection flow, inlet flue gas flow and inlet flue gas temperature showing strong correlation; indicating that there is high multiple correlation. For example, the flue gas flow can cause a change of the flue gas temperature, with a greater inlet flue gas flow resulting in a higher inlet flue gas temperature.

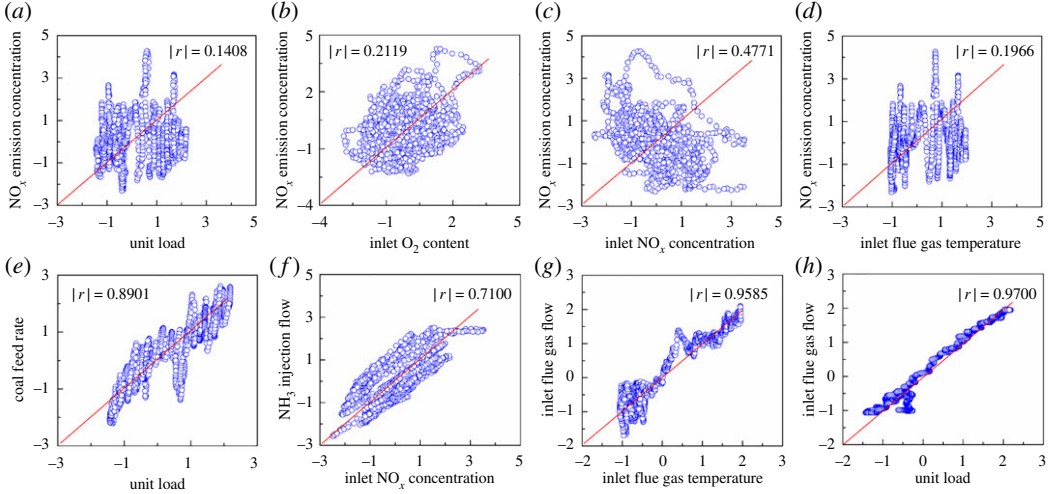

**Figure 4.** Correlation between the variables.

**Table 5.** The delay estimation result for each descriptor variable.

| descriptor variable | range of time delay | maximum MI | optimal result | time delay |
|---|---|---|---|---|
| $N_e$ | [20,60] | $MI(N_e, C_{NO_x,in}) = 0.9496$ | $\tau'_1 = 41, w'_1 = 116$ | $410 + 300 = 710$ s |
| $B$ | [20,60] | $MI(B, C_{NO_x,in}) = 1.0126$ | $\tau'_1 = 45, w'_1 = 56$ | $450 + 300 = 750$ s |
| $C_{NO_x,in}$ | [1,40] | $MI(C_{NO_x,in}, C_{NO_x,out}) = 1.1015$ | $\tau'_3 = 30, w'_3 = 84$ | $300$ s |
| $Q$ | [1,40] | $MI(Q, C_{NO_x,out}) = 1.1835$ | $\tau'_4 = 30, w'_4 = 84$ | $300$ s |
| $C_{O_2}$ | [1,40] | $MI(C_{O_2}, C_{NO_x,out}) = 0.1829$ | $\tau'_5 = 6, w'_5 = 105$ | $60$ s |
| $T$ | [1,40] | $MI(T, C_{NO_x,out}) = 0.5942$ | $\tau'_6 = 30, w'_6 = 120$ | $300$ s |
| $F$ | [1,40] | $MI(F, C_{NO_x,out}) = 0.8322$ | $\tau'_7 = 30, w'_7 = 120$ | $300$ s |

## 5.6. Analysis of knnMI-mwKPLS model parameters

The parameters of the knnMI-mwKPLS model include the wavelet kernel width $a_1$ and $a_2$, multi-scale parameter $c$ and the principal component $L$. Generally, $L$ and $c$ are selected as fixed values according to the sample characteristics. For further analysis, the following experimental data were chosen from time $t = 200$ min, and the sample size $n$ was 500.

First, multi-scale characteristic analysis of the training set was performed. FCM clustering was used to determine the scale $c$, and $W_c(U)$ was compared to obtain the optimal scale $c^*$.

It can be seen from table 6 that when $c = 2$, $W_c(U)$ reached the maximum and the clustering effect of the training set after FCM clustering was the best. Therefore, in this paper, $c = 2$ was used as the multi-scale parameter. Secondly, $L$ is determined by the leave-one-out cross-validation. The relationship between $L$ and $R_k^2(Y)$ is shown in figure 5.

It can be seen from figure 5 that when $k = 4$, the explained variance was $R_k^2(Y) \leq 0.0975$ and the total explained variance $R^2(Y)$ reached 93.17%. Noise would be included in the model if too many $L$ were extracted, which would affect the prediction accuracy; therefore, $L = 4$ was selected for this work.

To analyse the effects of different variables and phase space reconstruction on model performance, a training sample of $n = 500$ and a test sample of $nt = 200$ were used. The comparison results are shown in table 7.

(1) Dynamic modelling strategies often use an incrementally set variable to bring the system's dynamic characteristics into the model. For the mwKPLS model, if the descriptor variable only adds $x(t-1)$, the fitting accuracy and the prediction accuracy would both be reduced. When $y(t-1)$ is added, the fitting accuracy and the prediction accuracy both improved, similar results were obtained for the knnMI-mwKPLS model.

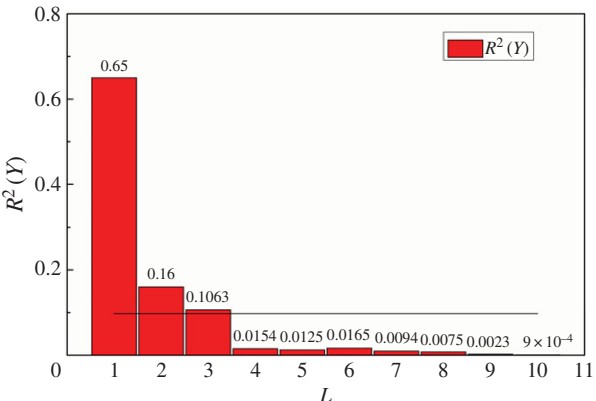

**Figure 5.** The relationship between the $L$ and $R_k^2(Y)$.

**Table 6.** Multi-scale characteristic analysis of the training set.

| sample $n$ | column $m$ | scale $c$ | fuzzy classification uncertainty $W_c(U)$ | optimal scale $c^*$ |
|---|---|---|---|---|
| 500 | 15 | 2 | 0.7419 | 2 |
| | | 3 | 0.7021 | |
| | | 4 | 0.6097 | |

**Table 7.** Performance comparison of the model with different variables and phase space reconstruction.

| | | | | RMSE | |
|---|---|---|---|---|---|
| model | variable | phase space reconstruction | dimension | training set (mg m$^{-3}$) | test set (mg m$^{-3}$) |
| mwKPLS | $x(t)$ | no | 7 | 0.8639 | 7.3568 |
| | $x(t-1), x(t)$ | | 14 | 1.5737 | 7.4873 |
| | $x(t-1), y(t-1), x(t)$ | | 15 | 1.2209 | 5.3755 |
| | $y(t-1), x(t)$ | | 8 | 0.9899 | 5.1095 |
| knnMI-mwKPLS | $x(t-\tau)$ | yes | 7 | 0.9723 | 8.2636 |
| | $x(t-\tau-1), x(t-\tau)$ | | 14 | 1.5164 | 8.9046 |
| | $x(t-\tau-1), y(t-1), x(t-\tau)$ | | 15 | 1.1542 | 5.1667 |
| | $y(t-1), x(t-\tau)$ | | 8 | 0.9171 | 4.8371 |

(2) The performance of the mwKPLS and knnMI-mwKPLS models were improved by adding the $y(t-1)$ variable, and the influence of phase space reconstruction was then further analysed. From the results in table 7, it can be verified that the fitting accuracy of the training set and the prediction accuracy on the test set could both be improved.

## 5.7. Dynamic inferential model analysis

In this paper, the dynamic model and the corrected dynamic model were analysed, and the model update performance was verified with the field data for different operating states, including the steady state, variable state and the blowback process of $NO_x$ analyser.

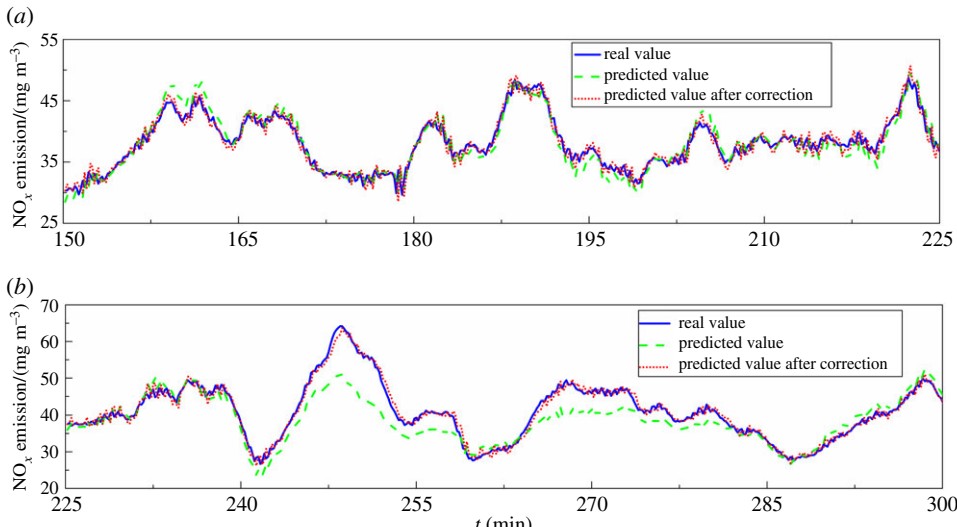

**Figure 6.** Real and predicted values after correction in the steady state and variable state.

**Table 8.** Comparison of the dynamic model and corrected dynamic model in the steady state and variable state.

| model | state | RMSE (mg m$^{-3}$) | $Q^2$ |
|---|---|---|---|
| dynamic model | steady | 1.4540 | 0.9038 |
| | variable | 4.4407 | 0.5343 |
| corrected dynamic model | steady | 1.1580 | 0.9289 |
| | variable | 1.2180 | 0.9739 |

The moving window method was used to select the steady-state samples and variable state from the preprocessed operating data. The steady-state determination criteria were evaluated using the stability factor (SF), which is shown by the below equation

$$\delta = \frac{x_{\max} - x_{\min}}{1/N \sum_{i=1}^{N} x_i} < \delta_0, \tag{5.4}$$

where $N$ is the length of the moving window, $x_{\max}$ and $x_{\min}$ are the maximal and minimal values, respectively, of samples ($x_i$, $i = 1, \ldots, N$) in the moving window and $\delta_0$ is the SF given previously. In this work, the boiler load was chosen as the feature variable for the state judgement. The $\delta_0$ was set to 0.083, $N$ was 200 and the sampling period was 10 s. Finally, the steady state and variable state samples were obtained according to the above criteria, as shown in figure 6.

It can be seen from figure 6a and table 8 that when the unit is in the steady state, the load is relatively stable. Therefore, the dynamic model showed good predictive accuracy with a low RMSE of 1.4540 mg m$^{-3}$ and a high coefficient of determination ($Q^2$) of 0.9038. The configuration parameters for the model include $L$, $c^*$, $a_1$ and $a_2$, with selected value of 4, 2, 4 and 20, respectively. After feedback correction, the corrected dynamic model showed slightly improved predictive ability. For the variable state, the boiler load gradually increased and a large amount of NO$_x$ was produced. The predictive ability of the dynamic model is lower at the peak of the NO$_x$ emission curve in figure 6b. At this time, the model configuration parameters were the same as those for the steady state. After feedback correction, the corrected dynamic model demonstrated a clear improvement in its predictive ability.

The NO$_x$ analyser periodically performs a blowback operation to ensure the cleanliness of the sampling system. At this time, the final measured value of the NO$_x$ emission concentration is maintained until the end of the blowback process; therefore, this is an important scenario for the dynamic inferential model.

The corrected dynamic model uses $y(t)$ for modelling and $y(t + 1)$ for correction; however, these values are not available at this point because $y(t)$ and $y(t + 1)$ are in the self-hold state during blowback. Therefore, it is necessary to substitute the predicted value $\hat{y}(t)$ for the real value $y(t)$ to calculate the predicted value $\hat{y}(t + 1)$ at time $t + 1$.

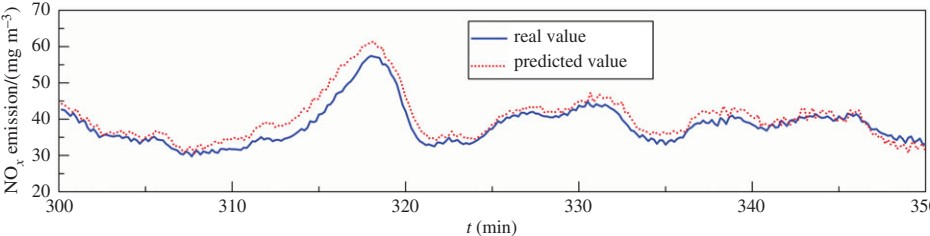

**Figure 7.** Predicted values using the dynamic model during the NO$_x$ analyser blowback process.

**Table 9.** Prediction accuracy of the dynamic model during the NO$_x$ analyser blowback process.

| model | RMSE | $Q^2$ |
|---|---|---|
| dynamic model | 2.6954 mg m$^{-3}$ | 0.8339 |

Assuming that the NO$_x$ analyser is in the blowback process from $t = 300$ min to $t = 350$ min and the model configuration parameters are $L = 4$, $c^* = 2$, $a_1 = 2$ and $a_2 = 18$, the results show that the dynamic model can still maintain high accuracy, as seen from figure 7 and table 9; the deviation between the predicted value and the real value is small, which effectively tracks the change in the curve, even for the highest or lowest points. When the NO$_x$ analyser reverts from the blowback process to normal operation, there is only a small disturbance to the model output.

In addition, the dynamic model also does not need frequent reconstruction and parameter updates, which is similar to the TD method. After analysis of numerous experimental results, it can be stated that even if the dynamic model adopts a different parameter, it has minimal effect on the accuracy of the model's predictions. Therefore, the dynamic model used a fixed parameter. The average time for model training was only 3.47 s for each update of the model, which meets engineering requirements.

## 6. Conclusion

In this paper, the multi-scale kernel and the Morlet wavelet kernel were combined to propose a new kernel function. The prediction accuracy of the mwKPLS model based on the new kernel function was further improved, as confirmed via verification using benchmark datasets.

Due to the response lag of the NO$_x$ analyser and the large inertia of the SCR reaction, the knnMI estimator could realize delay estimation and the model samples could be reconstructed. Therefore, the dynamic inferential model was able to accurately predict the NO$_x$ emissions one sampling period in advance.

In practice, abnormal operational condition of the boiler and the SCR system should be avoided; in particular, the continuous emission monitoring system should be in a normal work mode to ensure accuracy of the measured data. Under normal operating condition, the dynamic inferential model could better track the NO$_x$ emission trend under conditions with large fluctuation. If the deviation between the predicted value and the set value was large or the NO$_x$ analyser was in the blowback process, the NH$_3$ injection could be adjusted in time to adapt for load change, which is beneficial for improving the de-NO$_x$ efficiency and reducing NH$_3$ slip, which lays the foundation for design of the controller.

Data accessibility. This article does not contain any additional data.

Authors' contributions. L.Y. carried out the laboratory work, participated in data analysis, participated in the design of the study and drafted the manuscript. Z.D. conceived, designed and coordinated the study and helped draft the manuscript; J.H. and L.M. collected the field data; H.J. carried out the data analyses; all authors gave final approval for publication.

Competing interests. There are no competing interests to declare.

Funding. Financial support came from Natural Science Foundation of Hebei Province, China (grant no. E2018502111) and Fundamental Research Funds for the Central Universities, China (grant no. 2018QN097).

Acknowledgements. We are grateful to the two anonymous reviewers who provided comments that substantially improved the manuscript.

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
