## [Reviewer comments · Royal Society Open Science]

Review History

RSOS-191647.R0 (Original submission)

Review form: Reviewer 1

Is the manuscript scientifically sound in its present form?

Yes

Are the interpretations and conclusions justified by the results?

Yes

Is the language acceptable?

Yes

Do you have any ethical concerns with this paper?

No

Have you any concerns about statistical analyses in this paper?

No

Recommendation?

Accept with minor revision (please list in comments)

Comments to the Author(s)

Overall, this paper is well written with some minor mistakes, but the data is sufficient to support the conclusion, which indicated a better model for the NO_x emission.

1. Page 1, Line 34: Suggest rephrasing as: "The results proved that this dynamic model has shown...."
2. Page 1, Line 57: "so" should be changed to "however"
3. Page 2 line 5, give full definition of "SVM"
4. Format is not consistent in section 3.4. Line 27 is overlapped. Also, the format for figure titles are not consistent, for example, figure 6
5. Page 9 line 48: Delete "therefore"
6. Page 10 line 7-8, Suggest rephrasing as: "Those outliers are consistently mismatch with the baseline population, which adversely affect the statistical"
7. Page 10 line 10: Stable stabilize
8. In Figure 3, please comment on the correlation of all the parameters listed. Like: parameters a-d showed very weak correlations (no correlation) while e-h showing stronger correlation.
9. In conclusion, needs to mention the application condition limitation of this model, at which condition will be given the ideal projection.

Review form: Reviewer 2

Is the manuscript scientifically sound in its present form?

Yes

Are the interpretations and conclusions justified by the results?

Yes

Is the language acceptable?

No

Do you have any ethical concerns with this paper?

No

Have you any concerns about statistical analyses in this paper?

No

Recommendation?

Accept with minor revision (please list in comments)

Comments to the Author(s)

Overall I recommend that this paper be accepted with minor revisions. My concerns are listed within the attached document (Appendix A).

Decision letter (RSOS-191647.R0)

29-Nov-2019

Dear Dr Yan,

The editors assigned to your paper ("Dynamic Inferential NO_x Emission Prediction Model with Delay Estimation for SCR de-NO_x Process in Coal-fired Power Plants") have now received comments from reviewers. We would like you to revise your paper in accordance with the

referee and Associate Editor suggestions which can be found below (not including confidential reports to the Editor). Please note this decision does not guarantee eventual acceptance.

Please submit a copy of your revised paper before 22-Dec-2019. Please note that the revision deadline will expire at 00.00am on this date. If we do not hear from you within this time then it will be assumed that the paper has been withdrawn. In exceptional circumstances, extensions may be possible if agreed with the Editorial Office in advance. We do not allow multiple rounds of revision so we urge you to make every effort to fully address all of the comments at this stage. If deemed necessary by the Editors, your manuscript will be sent back to one or more of the original reviewers for assessment. If the original reviewers are not available, we may invite new reviewers.

- Data accessibility

<http://datadryad.org/submit?journalID=RSOS&manu=RSOS-191647>

- Competing interests

- Authors' contributions

All submissions, other than those with a single author, must include an Authors' Contributions section which individually lists the specific contribution of each author. The list of Authors should meet all of the following criteria; 1) substantial contributions to conception and design, or

acquisition of data, or analysis and interpretation of data; 2) drafting the article or revising it critically for important intellectual content; and 3) final approval of the version to be published.

- Acknowledgements

- Funding statement

Kind regards,
Lianne Parkhouse
Editorial Coordinator
Royal Society Open Science
openscience@royalsociety.org

on behalf of Professor Kerry Rowe (Subject Editor)
openscience@royalsociety.org

Associate Editor's comments:

Thank you for submitting this manuscript to Royal Society Open Science. You will note that the two reviewers have provided a number of comments regarding corrections that are necessary to improve the qualitative standard of the paper. Not only are scientific/technical concerns raised but also concerns regarding the presentation, specifically the quality of the written English and a number of the figures included with the paper. Please ensure you enhance the quality of the figures and seek English language editing assistance - for example, <https://royalsociety.org/journals/authors/language-polishing/>.

As the reviewers have requested these improvements, you are expected to fully respond to and incorporate them into your revision - if you choose not to do so, the Editors reserve the right to reject the paper from further consideration. This is in line with our general policy of only permitting one round of major revision under most circumstances.

Reviewers' Comments to Author:

Reviewer: 1

Comments to the Author(s)

Overall, this paper is well written with some minor mistakes, but the data is sufficient to support the conclusion, which indicated a better model for the NO_x emission.

1. Page 1, Line 34: Suggest rephrasing as: "The results proved that this dynamic model has shown...."
2. Page 1, Line 57: "so" should be changed to "however"
3. Page 2 line 5, give full definition of "SVM"
4. Format is not consistent in section 3.4. Line 27 is overlapped. Also, the format for figure titles are not consistent, for example, figure 6
5. Page 9 line 48: Delete "therefore"
6. Page 10 line 7-8, Suggest rephrasing as: "Those outliers are consistently mismatch with the baseline population, which adversely affect the statistical"
7. Page 10 line 10: Stable stabilize
8. In Figure 3, please comment on the correlation of all the parameters listed. Like: parameters a-d showed very weak correlations (no correlation) while e-h showing stronger correlation.
9. In conclusion, needs to mention the application condition limitation of this model, at which condition will be given the ideal projection.

Reviewer: 2

Comments to the Author(s)

Overall I recommend that this paper be accepted with minor revisions. My concerns are listed within the attached document.

Author's Response to Decision Letter for (RSOS-191647.R0)

See Appendices B - C.

RSOS-191647.R1 (Revision)

Review form: Reviewer 1

Is the manuscript scientifically sound in its present form?

Yes

Are the interpretations and conclusions justified by the results?

Yes

Is the language acceptable?

Yes

Do you have any ethical concerns with this paper?

No

Have you any concerns about statistical analyses in this paper?

No

Recommendation?

Accept as is

Comments to the Author(s)

Thank you for the feedback, Suggest to accept.

Review form: Reviewer 2

Is the manuscript scientifically sound in its present form?

Yes

Are the interpretations and conclusions justified by the results?

Yes

Is the language acceptable?

Yes

Do you have any ethical concerns with this paper?

No

Have you any concerns about statistical analyses in this paper?

No

Recommendation?

Accept as is

Comments to the Author(s)

The revisions that have been made according to each comment are acceptable. Thank you.

Decision letter (RSOS-191647.R1)

08-Jan-2020

Dear Dr Yan,

It is a pleasure to accept your manuscript entitled "Dynamic inferential NO_x emission prediction model with delay estimation for SCR de-NO_x process in coal-fired power plants" in its current form for publication in Royal Society Open Science. The comments of the reviewer(s) who reviewed your manuscript are included at the foot of this letter.

on behalf of Prof R. Kerry Rowe (Subject Editor)
openscience@royalsociety.org

Associate Editor Comments to Author:

Thank you for providing a response to the reviewers and incorporating their recommendations. The referees are now satisfied your manuscript may be published in its current form. The editorial and production offices will be in touch in due course.

Reviewer comments to Author:

Reviewer: 1

Comments to the Author(s)

Thank you for the feedback, Suggest to accept.

Reviewer: 2

Comments to the Author(s)

The revisions that have been made according to each comment are acceptable. Thank you.

Appendix A

Journal: Royal Society Open Science

Manuscript No: RSOS-191647

Title: Dynamic Inferential NO_x Emission Prediction Model with Delay Estimation for SCR de-NO_x Process in Coal-fired Power Plants

Document : Comments to the Author(s)

- The manuscript contains numerous grammatical and sentence structure errors in the English language. The authors should revise the manuscript to address these issues and it is recommended that the authors have a native English speaker work through the manuscript with them to fully resolve such errors.
- Literature review of data-driven methods applied to NO_x formation from coal-fired power plants is lacking. Please consider “S. M. Safdarnejad, J. F. Tuttle, and K. M. Powell, “Dynamic modeling and optimization of a coal-fired utility boiler to forecast and minimize NO_x and CO emissions simultaneously,” *Comput. Chem. Eng.*, vol. 124, pp. 62–79, 2019.” and “J. F. Tuttle, R. Vesel, S. Alagarsamy, L. D. Blackburn, and K. Powell, “Sustainable NO_x emission reduction at a coal-fired power station through the use of online neural network modeling and particle swarm optimization,” *Control Eng. Pract.*, vol. 93, no. September, 2019.”
- References to equations within the text should be done by referring to the defined equation numbers. Please address instances where this is not done for clarity.
- The reported total coal feed rate is not adequate to realize the stated generation level (700+ MW). Other unit parameters reported in Table 4 are also suspect (inlet flue gas flow rate, NH₃ injection flow rate). Are these being reported correctly?
- Quality of figures is poor (particularly Figures 2 & 5). Please enhance.
- A Table of Nomenclature should be included describing each variable throughout the mathematical equation, as well as each acronym.
- The identifying method used to determine from the dataset unit “variable state” vs. “steady state” should be reported, and more information provided of the identity of each dataset (e.g. number of records, ramping rates of variable state, etc.)
- What was the training procedure for the NO_x model used in the field data experiment? Was the reported dataset of 2100 samples separated into training and test data? Was other data used for training and the 2100 sample set strictly test data? Table 7 shows results of determining optimal phase space reconstruction with train/test data of 500/200 records, however it is unclear if it is one of these models or a separately trained model that is used to generate the final results (Figure 5). The final model configuration and all selected parameters for each analysis step/method should be summarized within the final results discussion.

Journal: Royal Society Open Science

Manuscript No: RSOS-191647

Title: Dynamic Inferential NO_x Emission Prediction Model with Delay Estimation for SCR de-NO_x Process in Coal-fired Power Plants

Document : Comments to the Author(s)

- A schematic view outlining the algorithm steps described on page 7 would contribute to clarity. Particularly, show the interaction of data and results between each step.

Appendix B

Dear Reviewer1:

Thank you for the reviewers' comments concerning our manuscript entitled "Dynamic Inferential NO_x Emission Prediction Model with Delay Estimation for SCR de-NO_x Process in Coal-fired Power Plants"(ID: RSOS-191647). Those comments are all valuable and very helpful for revising and improving our paper, as well as the important guiding significance to our researches. We have studied comments carefully and have made correction which we hope meet with approval. Revised portions are marked in red in the paper. The main corrections in the paper and the responds to the reviewer's comments are as flowing:

Responds to the reviewer's comments:

Reviewer #1:

1. Response to comment: (Page 1, Line 34: Suggest rephrasing as: "The results proved that this dynamic model has shown....")

Response: Thank you for your suggestions. The statement of "the results show: the dynamic inferential model has shown...." was corrected as "the results proved that this dynamic model has"

2. Response to comment: (Page 1, Line 57: "so" should be changed to "however")

Response: Thank you for your suggestions. The statement of "So it cannot correctly reflect...." was corrected as "However, it would not be able to...."

3. Response to comment: (Page 2 line 5, give full definition of "SVM")

Response: Thank you for your suggestions. The statement of "Bao et al. used the multi-scale kernel to improve the prediction ability of SVM" was corrected as "Bao et al. used the multi-scale kernel to improve the prediction ability of support vector machine(SVM)."

4. Response to comment: (Format is not consistence in section 3.4. Line 27 is overlapped. Also, the format for figure titles are not consistent, for example, figure 6)

Response: We are very sorry for our incorrect writing. The problem which Line 27 in section 3.4 is overlapped has corrected. The problem is due to the use of MathType software. The equation is normal in the Word file, but there is a problem when it is converted into a PDF file. The formats for figure titles were all checked and made consistently.

5. Response to comment: (Page 9 line 48: Delete "therefore")

Response: Thank you for your suggestions. "therefore" was deleted in line 48 Page 9.

6. Response to comment: (Page 10 line 7-8, Suggest rephrasing as: "Those outliers are consistently mismatch with the baseline population, which adversely affect the statistical")

Response: Thank you for your suggestions. The statement of "These outliers are significantly inconsistent with the trend of the surrounding data, which adversely affect the statistical....." was corrected as "Those outliers were consistently mismatched with the baseline population, which adversely affected the statistical"

7. Response to comment: (stabilize◊Page 10 line 10: Stable)

Response: Thank you for your suggestions. The statement of “....., which does not help to stable modeling.” was corrected as “....., which does not help stabilize the model.”

8. Response to comment: (In Figure 3, please comment on the correlation of all the parameters listed. Like: parameters a-d showed very weak correlations (no correlation) while e-h showing stronger correlation.)

Response: As Reviewer suggested that the correlation of all the parameters listed were needed to be commented. So, the comments which added in Figure 3 were as follows:

“The unit load, inlet O₂ content, inlet flue gas temperature, and NO_x emission concentration display very weak correlations while the inlet NO_x concentration and NO_x emission concentration show a slightly stronger correlation..... including the coal feed rate, inlet flue gas flow and unit load, inlet NO_x concentration and NH₃ injection flow, inlet flue gas flow and inlet flue gas temperature showing strong correlation;”

9. Response to comment: (In conclusion, needs to mention the application condition limitation of this model, at which condition will be given the ideal projection.)

Response: As reviewer suggested that the application condition limitation of this model and the condition which could give the ideal projection were needed to be mentioned. So, the contents added in the conclusion were as follows:

“In practice, abnormal operational condition of the boiler and the SCR system should be avoided, in particular, the continuous emission monitoring system (CEMS) should be in a normal work mode to ensure accuracy of the measured data. Under normal operating condition, the dynamic inferential model could better track the NO_x emission trend under conditions with large fluctuation.”

Special thanks to you for your good comments.

In all, I found your comments are quite helpful, and I revised my paper point-by-point. And we tried our best to improve the manuscript and made some changes in the manuscript according to the English language editing assistance. These changes will not influence the content and framework of the paper. And here we did not list the changes but marked in red in revised paper.

We appreciate for your warm work earnestly, and hope that the correction will meet with approval.

Once again, thank you very much for your comments and suggestions, and I feel so sorry that so much of your precious time was wasted on our paper revision.

Appendix C

Dear Reviewer 2:

Thank you for the reviewers' comments concerning our manuscript entitled "Dynamic Inferential NO_x Emission Prediction Model with Delay Estimation for SCR de-NO_x Process in Coal-fired Power Plants"(ID: RSOS-191647). Those comments are all valuable and very helpful for revising and improving our paper, as well as the important guiding significance to our researches. We have studied comments carefully and have made correction which we hope meet with approval. Revised portions are marked in red in the paper. The main corrections in the paper and the responds to the reviewer's comments are as flowing:

Reviewer #2:

1. Response to comment: (The manuscript contains numerous grammatical and sentence structure errors in the English language. The authors should revise the manuscript to address these issues and it is recommended that the authors have a native English speaker work through the manuscript with them to fully resolve such errors.)

Response: Thank you for your suggestions. I have revised the manuscript and sought English language editing assistance.

2. Response to comment: (Literature review of data-driven methods applied to NO_x formation from coal-fired power plants is lacking. Please consider "S. M. Safdarnejad, J. F. Tuttle, and K. M. Powell, "Dynamic modeling and optimization of a coal-fired utility boiler to forecast and minimize NO_x and CO emissions simultaneously," *Comput. Chem. Eng.*, vol. 124, pp. 62–79, 2019." and "J. F. Tuttle, R. Vesel, S. Alagarsamy, L. D. Blackburn, and K. Powell, Sustainable NO_x emission reduction at a coal-fired power station through the use of online neural network modeling and particle swarm optimization," *Control Eng. Pract.*, vol. 93, no. September, 2019.")

Response: Thank you for your suggestions. I have added the two mentioned literature in the literature review. So, the added contents were as follows:

"Safdarnejad et al. developed a data-driven model based on the recurrent NN model and the dynamic particle swarm optimizer to simultaneously estimate NO_x and CO emissions [5]. Tuttle et al. presented a unique NN model utilizing swappable synapse weights and the hybrid optimization approach in a combustion optimization system [6]."

The two literatures were cited as the reference [5] and reference [6].

[5]Safdarnejad S M, Tuttle J F, Powell K M. 2019 Dynamic modeling and optimization of a coal-fired utility boiler to forecast and minimize NO_x and CO emissions simultaneously. *Comput.Chem.Eng.* **124**, 62-79. (doi: 10.1016/j.compchemeng.2019.02.001)

[6]Tuttle J F, Vesel R, Alagarsamy S, Blackburn L D, Powell K. 2019 Sustainable NO_x emission reduction at a coal-fired power station through the use of online neural network modeling and particle swarm optimization. *Control Eng.Pract.* **93**, 104167. (doi: 10.1016/j.conengprac.2019.104167)

3. Response to comment: (References to equations within the text should be done by referring to the defined equation numbers. Please address instances where this is not done for clarity)

Response: We are very sorry for our incorrect writing. The equation numbers include

equation (2.2), equation (2.4) to equation (2.5), equation (2.7), equation (2.10), equation (3.2) , equation (3.6), equation (3.10), equation (3.14) to equation (3.16), equation (3.16), equation (3.23) to equation (3.24), and equation (5.1) to equation (5.4) were newly defined. References to equation were done by the defined equation numbers. In addition, all the other equation numbers were all corrected, especially in section2, section 3 and section 5.

4. The reported total coal feed rate is not adequate to realize the stated generation level (700+MW). Other unit parameters reported in Table 4 are also suspect (inlet flue gas flow rate, NH₃ injection flow rate). Are these being reported correctly?

Response: We are very sorry for our incorrect writing. The unit of total coal feed rate “kg/h” was corrected as “t/h”. The unit of inlet flue gas flow rate “m³/h” was corrected as “km³/h”. The unit of NH₃ injection flow rate “m³/h” was corrected as “kg/h”.

5. Response to comment: (Quality of figures is poor (particularly Figures 2 & 5). Please enhance.)

Response: Thank you for your suggestions. The qualities of figures (from Figure 2 to Figure 6) were all improved.

6. Response to comment: (A Table of Nomenclature should be included describing each variable throughout the mathematical equation, as well as each acronym.)

Response: Thank you for your suggestions. Nomenclatures of all variables were added in Table 4. In addition, Table 5 was corrected.

7. Response to comment: (The identifying method used to determine from the dataset unit “variable state” vs. “steady state” should be reported, and more information provided of the identity of each dataset (e.g. number of records, ramping rates of variable state, etc.)

Response: Thank you for your suggestions. I have modified it in accordance with your suggestions. So, the added contents were as follows:

“The moving window method was used to select the steady state samples and variable state from the pre-processed operating data. The steady state determination criteria were evaluated using the stability factor (SF), which is shown by Equation (5.4).

$$\delta = \frac{x_{\max} - x_{\min}}{\frac{1}{N} \sum_{i=1}^N x_i} < \delta_0 \quad (5.4)$$

where N is the length of the moving window, x_{\max} and x_{\min} are the maximal and minimal values, respectively, of samples $(x_i, i = 1, \dots, N)$ in the moving window, and δ_0 is the SF given previously. In this work, the boiler load was chosen as the feature variable for the state judgment. The δ_0 was set to 0.083, N was 200, and the sampling period was 10 s. Finally, the steady state and variable state samples were obtained according to the above criteria, as shown in fig. 6.”

8. Response to comment: (What was the training procedure for the NO_x model used in the field data experiment? Was the reported dataset of 2100 samples separated into training and test data? Was other data used for training and the 2100 sample set strictly test data? Table 7 shows

results of determining optimal phase space reconstruction with train/test data of 500/200 records, however it is unclear if it is one of these models or a separately trained model that is used to generate the final results (Figure 5). The final model configuration and all selected parameters for each analysis step/method should be summarized within the final results discussion.)

(1) What was the training procedure for the NO_x model used in the field data experiment?

Response: In this paper, the preprocessed model samples were arranged in time series. Then, phase space reconstruction was performed on the preprocessed samples based on the time delay results. The total number of selected modeling samples was 700. So, the first 500 samples were the training set, and the last 200 samples were the test set. When a new sample was acquired, the model samples were updated. The new sample was added to the test set samples, and the oldest sample was deleted from the training set. So, when the model samples were dynamic updated each time, the response variable for the next sampling time was predicted in advance.

(2) Was the reported dataset of 2100 samples separated into training and test data?

Response: As mentioned above, the total number of selected modeling samples was 700. So, the first 500 samples were the training set, and the last 200 samples were the test set. So, some of 2100 samples were separated into training and test data.

(3) Was other data used for training and the 2100 sample set strictly test data?

Response: The predicted value curve in Figure 5 was a combination of the predicted values which outputted by the dynamic model each time when model samples were updated. As mentioned above, some of 2100 samples were separated into training and test data.

(4) Table 7 shows results of determining optimal phase space reconstruction with train/test data of 500/200 records, however it is unclear if it is one of these models or a separately trained model that is used to generate the final results (Figure 5)

Response: Due to changes in operating conditions, the results of delay estimation may be different. So, it is one of these models which are used to generate the final results in Figure 5.

(5) The final model configuration and all selected parameters for each analysis step/method should be summarized within the final results discussion.

Response: Thank you for your suggestions. So, the added contents were as follows:

“..... , that when the unit is in the steady state, The configuration parameters for the model include L , c^* , a_1 , and a_2 , with selected value of 4, 2, 4, and 20, respectively.”

“For variable state, At this time, the model configuration parameters were the same as those for the steady state.”

“.....NO_x analyzer is in the blowback process from $t=300$ min to $t=350$ min and the model configuration parameters are $L=4$, $c^*=2$, $a_1=2$, and $a_2=18$, the results show that.....”

9. Response to comment: (A schematic view outlining the algorithm steps described on page 7 would contribute to clarity. Particularly, show the interaction of data and results between each step.)

Response: Thank you for your suggestions. A schematic view which outlined the algorithm steps was added as Figure 1.

Special thanks to you for your good comments.

In all, I found your comments are quite helpful, and I revised my paper point-by-point. And we tried our best to improve the manuscript and made some changes in the manuscript according to the English language editing assistance. These changes will not influence the content and framework of the paper. And here we did not list the changes but marked in red in revised paper.

We appreciate for your warm work earnestly, it's obvious that you are an expert, and there is lots of shortage in our study which need your instruction.

Once again, thank you very much for your comments and suggestions, and I feel so sorry that so much of your precious time was wasted on our paper revision.